# Effective Biocorrosive Control in Oil Industry Facilities: 16S rRNA Gene Metabarcoding for Monitoring Microbial Communities in Produced Water

**DOI:** 10.3390/microorganisms11040846

**Published:** 2023-03-27

**Authors:** Joyce Dutra, Glen García, Rosimeire Gomes, Mariana Cardoso, Árley Côrtes, Tales Silva, Luís de Jesus, Luciano Rodrigues, Andria Freitas, Vinicius Waldow, Juliana Laguna, Gabriela Campos, Monique Américo, Rubens Akamine, Maíra de Sousa, Claudia Groposo, Henrique Figueiredo, Vasco Azevedo, Aristóteles Góes-Neto

**Affiliations:** 1Department of Microbiology, Institute of Biological Sciences, Federal University of Minas Gerais, Belo Horizonte 31270-901, MG, Brazil; dutra.engenharia.ambiental@gmail.com (J.D.); rosi.floripes@gmail.com (R.G.); vasco@icb.ufmg.br (V.A.); 2Department of Genetics Ecology and Evolution, Institute of Biological Sciences, Federal University of Minas Gerais, Belo Horizonte 31270-901, MG, Brazil; arleyrezcort@gmail.com (Á.C.); talesfdasilva@gmail.com (T.S.); luiislimma@gmail.com (L.d.J.); andria.sfreitas@gmail.com (A.F.); jujulaguna@gmail.com (J.L.); gabrielamunis24@gmail.com (G.C.); moniquefamerico@gmail.com (M.A.); 3Departments of Bioinformatic, Institute of Biological Sciences, Federal University of Minas Gerais, Belo Horizonte 31270-901, MG, Brazil; glen.yupanqui@gmail.com (G.G.); marianascardoso@yahoo.com.br (M.C.); 4Department of Veterinary Medicine, Faculty of Veterinary, Federal University of Minas Gerais, Belo Horizonte 31270-901, MG, Brazil; lsantosrodrigues@gmail.com (L.R.); figueiredoh@icb.ufmg.br (H.F.); 5Petrobras Research and Development Center (CENPES), Petrobras, Rio de Janeiro 21941-915, RJ, Brazil; vinicius.waldow@petrobras.com.br (V.W.); akamine@petrobras.com.br (R.A.); mpsousa@petrobras.com.br (M.d.S.); claudiagroposo@gmail.com (C.G.)

**Keywords:** petroleum, produced water, microbiologically influenced corrosion, oil industry, metabarcoding

## Abstract

Microbiologically influenced corrosion (MIC) or biocorrosion is a complex biological and physicochemical process, Strategies for monitoring MIC are frequently based on microbial cultivation methods, while microbiological molecular methods (MMM) are not well-established in the oil industry in Brazil. Thus, there is a high demand for the development of effective protocols for monitoring biocorrosion with MMM. The main aim of our study was to analyze the physico-chemi- cal features of microbial communities occurring in produced water (PW) and in enrichment cultures in oil pipelines of the petroleum industry. In order to obtain strictly comparable results, the same samples were used for both culturing and metabarcoding. PW samples displayed higher phylogenetic diversity of bacteria and archaea whereas PW enrichments cultures showed higher dominance of bacterial MIC-associated genera. All samples had a core community composed of 19 distinct genera, with MIC-associated *Desulfovibrio* as the dominant genus. We observed significant associations between the PW and cultured PW samples, with a greater number of associations found between the cultured sulfate-reducing bacteria (SRB) samples and the uncultured PW samples. When evaluating the correlation between the physicochemical characteristics of the environment and the microbiota of the uncultivated samples, we suggest that the occurrence of anaerobic digestion metabolism can be characterized by well-defined phases. Therefore, the detection of microorganisms in uncultured PW by metabarcoding, along with physi-cochemical characterization, can be a more efficient method compared to the culturing method, as it is a less laborious and cost-effective method for monitoring MIC microbial agents in oil industry facilities.

## 1. Introduction

Microbiologically influenced corrosion (MIC) or biocorrosion is a process the leads to the deterioration of pipelines, tanks, and other facilities in the oil industry due to the metabolic activities of microorganisms [1,2]. These microorganisms mainly include Archaea and Bacteria, but also eukaryotes, such as Fungi [3,4,5]. Fungal community and aerobic prokaryotes consume molecular oxygen (O_2_), contributing to the establishment and survival of anaerobic microorganisms, which may have mechanisms that trigger biocorrosion, such as the reduction of Fe(0) through extracellular electron transfer (EET-MIC) [6].

In addition, some sulfate-reducing archaea (SRA), such as *Archaeoglobus*, contributes to a significant increase in corrosion due to the reduction of sulfate (SO_4_^2−^) to hy-drogen sulfide (H_2_S) [7]. Nonetheless, bacteria are usually the most studied microorganisms associated with the process of metallic biocorrosion in the oil industry [8]. Sulfate-reducing bacteria (SRB) and acid-producing bacteria (APB) produce H_2_S and organic acids, which are highly corrosive products, as an integral part of their metabolism [4,8]. Both ecological groups are classic examples of microorganisms that contribute to MIC in oil industries [4,8,9,10]. Other bacterial groups that deserve attention in biocorrosion processes are elemental sulfur-reducing bacteria (S_0_RB) [11], iron-reducing bacteria (IRB) [12], nitrate-reducing bacteria (NRB) [13], and thiosulfate-reducing bacteria (TRB) [14]. 

MIC monitoring is a complex task due to the heterogeneity of biofilms composition and distribution inside pipelines, as well as the localized nature of biocorrosion [15,16]. In oil industry facilities, microorganisms are distributed in different environments [17], either as biofilm components adhered to the surfaces of pipelines and tanks [1,18], in water microdroplets dispersed in the oil phase [17], or suspended in the aqueous phase [19]. The aqueous phase can be composed of formation water (FW), originally present in the reservoir, injection water (IW), used for enhanced oil recovery [19], and produced water (PW), that can be a mixture of the FW and IW [20]. 

Metallic installations in the oil industry are prone to failures [2], and these failures are often associated with corrosion processes: about USD 2.5 trillion is spent yearly on repairs and mitigation of its consequences [21]. Among that total, it is estimated that 10 to 20% of those cases involve MIC [21,22,23]. The culture-dependent most probable number (MPN) method based on serial dilution in culture media is the most widely used method for the detection and quantification of microorganisms in the oil industry [24], and it is commonly used to monitor SRB, APB, and general anaerobic heterotrophic bacteria (GANB) [25]. 

In recent decades, some studies have shown that monitoring using MPN for the purposes of MIC risk assessment can provide inconclusive and incorrect results, leading to erroneous actions [26,27]. This is because, usually, no more than 1% of microbial species present in environmental samples are able to grow in a specific culture medium [26,27,28]. 

Culture-independent techniques such as microbiological molecular methods (MMM) have been attracting the interest of oil industry operators [21,29,30]. The application of MMM has made it possible to more comprehensively and accurately detect and identify microbial groups and species occurring in complex substrates [31]. Those methods rely on biomarkers such as adenosine triphosphate (ATP), ribonucleic acid (RNA), deoxyribonucleic acid (DNA), proteins, and metabolites [32,33]. 

Metagenomics is based on the large-scale sequencing of DNA molecules directly extracted from environmental samples [34], which allows for a genomic investigation of microbial communities without the need for prior cultivation in the laboratory [35]. Thus, Next-Generation Sequencing (NGS) technology associated with the development of massively parallelized sequencing platforms and bioinformatic programs to analyze generated data have promoted an exponential increase in the number of genomes and metagenomes sequenced at low cost, contributing to the dissemination and improvement of metagenomics [10,36,37]. Furthermore, metagenomics can employ a pre-amplification step followed by large-scale sequencing of genomic regions of biomarkers, an approach called amplicon metagenomics or metabarcoding [21].

In this study, we performed metabarcoding to analyze the spatial variation in the structure of bacterial and archaeal communities in PW from storage tanks and enrichments obtained on the culture media for SRB, APB, and GANB, and their correlation with physico-chemical features of PW.

## 2. Materials and Methods

### 2.1. Sampling of Produced Water (PW)

The collection of PW samples was performed in an oil storage and transfer terminal, located at Duque de Caxias-RJ, Brazil. PW samples were collected from side valves connected at two heights (1.00 m and 2.75 m) of a drainer tank, a total of 14.63 m, which receives the residual volume of water and oil from other storage tanks. Sampling was performed in triplicate using three sterile plastic containers of 5.0 L per sampling point. The samples were immediately transported at room temperature, for a maximum of 2 h, to the laboratory, where all the samples were subjected to three distinct processes: (i) physicochemical characterization (lactate, acetate, propionate, formate, butyrate, sulfate (SO_4_^2−^), soluble sulfides (S^2−^), and chloride (Cl^−^) (ii) culturing by inoculation in distinct flasks containing specific culture media for the growth of SRB, APB, and GANB; (iii) filtering with the use of membranes (Kasvi, Curitiba, Brazil) with 0.22 μm diameter mesh using vacuum pumps for microbial cell retention and subsequent extraction of metagenomic DNA.

The fatty acid and ions analysis was perfomed using a Dionex-2500 ion chromatograph (Dionex Corporation, Sunnyvale, CA, USA) with an AS18 (4 × 250 mm) column at 30 °C with a flow rate of 1.0 mL/min for 30 min per sample eluted in a 2–40 mmol/L gradient of KOH [38,39]. The alkalinity was analyzed by potentiometric titration [40]. The Agilent SpectrAA 280 atomic absorption spectrometer was used for the quantification of Fe. The flame used was air acetylene, with a wavelength of 248.3 nm and a slit width of 0.2 nm. The calibration curve was prepared from a 1000 mg/L iron standard solution. In this protocol it is not possible to distinguish between Fe^2+^ and Fe^3+.^ The pH and temperature were analyzed using a Kasvi pH-meter (K39-1014B).

### 2.2. Produced Water (PW)

Microbial cells from PW samples from collection points at 1.00 m and 2.75 m were concentrated using disposable filtration systems with polyethersulfone (PES) membrane filter (91 mm diameter and 0.22 µm pore size). From each sampling point, approximately 450 mL of PW were filtered. Filters (Kasvi, Curitiba, Brazil) with retained microbial cells were placed in sterile 50 mL tubes and stored in an ultrafreezer (−80 °C) for subsequent extraction of DNA.

### 2.3. SRB, APB, and GANB Enrichment Cultures Obtained from PW

Detection and quantification of microorganisms of major physiological groups (SRB, APB, and GANB) were performed using the MPN method [41]. The MPNs of acid-producing bacteria (APB), sulfate-reducing bacteria (SRB), and general anaerobic heterotrophic bacteria (GANB) were determined from ten-fold serial dilutions ranging from 10^−0^ to 10^−8^ using 1 mL of PW samples (of 1.00 m and 2.75 m) inoculated into 10 mL flasks containing 9 mL specific medium for each group (APB, SRB, and GANB), and the time between collection and testing did not exceed 48 h. The assay was performed in triplicate. The SRB culture medium composition was: 0.5 g/L KH_2_PO_4_, 1 g/L NH_4_Cl, 1 g/L Na_2_SO_4_, 1 g/L CaCl_2_·6H_2_O, 1.83 g/L MgCl_2_·6H_2_O, 1 g/L yeast extract, 7 ml/L sodium lactate, 0.5 g/L FeSO_4_·7H_2_O, 1.9 g/L agar-agar, 4 mL/L rezasurin, and 0.12 g/L sodium thioglycolate, with a pH of 7.6 [42]. For the culturing of APB, the culture medium TSI (Agar Triple Sugar Iron) was used composition: 3 g/L meat extract, 3.g/L yeast extract, 15 g/Lcasein pancreatic digest, 5 g/L protease peptone, 1 g/L dextrose, 10 g/L lactose, 10 g/L sucrose, 0.2 g/L ferrous sulfate, 5 g/L sodium chloride, 0.3 g/L sodium thiosulfate, 12 g/L agar, and 24 mg phenol red [43]. In turn, for the culturing of GANB, the medium with the following composition was used: 5 g/L glucose, 4 g/L universal peptone, 1 g/Lyeast extract and 4.0 mL of rezasurin solution 0.025% *m/v* [44].

The flasks containing the cultures were incubated at a temperature of 30 ± 2 °C (similar to in situ temperature), for a total of 28 days. After incubation, the first flask inoculated with PW was selected for further DNA extraction. Cultures were centrifuged at 11,000× *g* for 1 min at room temperature. For quantification, mathematical equations were used, based on Poisson’s law, as shown in the Appendix A. Cell pellets were stored in an ultrafreezer (−80 °C) for later extraction of DNA. The first flask inoculated with PW was selected for further DNA extraction.

### 2.4. DNA Extraction

The DNA from PW was extracted from membrane filters with 100 and 250 mg retained microbial cells from PW (p_1.00 and p_2.75 m) and from 0.1 g of pellet from SRB, APB, and GANB enrichment cultures obtained from PW (from points 1.00 and 2.75 m). DNA extraction was performed in triplicate using the FastDNA™ Spin Kit for Soil (MP Biomedicals, Jacksonville, USA), following the manufacturer’s instructions. Quantification and purity (evaluated through A260/280 and A260/230) were confirmed in the NanoDrop (Thermo Fisher Scientific, Waltman, USA). DNA integrity was verified through visualization of 1.0% (*m/v*) agarose gel electrophoresis.

### 2.5. 16S rRNA Gene Amplicon Metagenomic Sequencing

Amplification was performed with primers for the V_3_–V_4_ region of the 16S rRNA gene: 341F (5′-CCTACGGGRSGCAGCAG-3′) [45] and 806R (5′-GGACTACHVGGGTWTCTAAT-3′) [46]. Library preparation and sequencing were performed in the MiSeq sequencing platform (Illumina Inc., USA), using the paired-end type Kit V2 for 500 cycles (2 × 250 bp).

### 2.6. Bioinformatic Analyses

Metabarcoding data were analyzed using the Amplicon Sequence Variant (ASV) approach and utilizing a customized pipeline developed in Python 3. The pipeline source code is deposited on GitHub (https://github.com/LBMCF/pipeline-for-amplicon-analysis), accessed on 14 January 2022.

This pipeline uses USEARCH [47] of 32-bits, VSEARCH [48], Cutadapt [49], and FastQC [50]. Initially, the quality control of the reads was evaluated by FastQC. Then, all reads were merged with USEARCH using the —fastq_merge pairs option, and sequencing adapters were cut with Cutadapt. Quality control was performed with VSEARCH using the —fastq_filter and —fastq_maxee = 0.8 options. For the elimination of duplicates, VSEARCH was also used with the option —derep_fulllength. The ASV table was generated with USEARCH with the —unoise3 option. The removal of chimeras and the generation of the table of ASVs were performed using the USEARCH unoise3 algorithm (option —unoise3). Additionally, the generation of the absolute abundances table was performed with the usearch_global and id = 0.99 VSEARCH options. The taxonomic identification and classification of ASVs were performed using SINTAX [51] in USEARCH with the database SILVA SSU NR 138.1 [52] as a reference. Finally, the consolidation of the abundance table with the taxonomic identifications was carried out with the customized script get_abundances_table_asv.py, included in the source code. For ecological analyses, the taxonomic level of genus was used.

The quality of the sequencing of the third replicate of the APB_1.00 sample was not satisfactory. Therefore, a statistical inference strategy was used, in which the ASVs of the third replicate of the variable (APB_1.00) were inferred from the first and second replicate. This technique is called the conditional mean algorithm. This technique was used to maintain the standard of three replicates per sample. The rules used are presented below:If x = 0 and y = 0, then the conditional mean is 0;If x = 5 and y = 0, then the conditional mean is 5;If x = 0 and y = 8, then the conditional mean is 8;If x = 3 and y = 4, then the conditional mean is 3.5 (or 3, since the integer part is used);If x = 6 and y = 9, then the conditional mean is 7.5 (or 7, since the integer part is used).

### 2.7. Integrating Metabarcoding and of PW Samples and Enrichments 

Relative abundances (%) were calculated from the ASVs and analyses were performed at the phylum and genus levels. These relative abundances were calculated using the dplyr package (v1.1.0) in R [53]. Comparative analyses of PW metabarcoding data and its corresponding enrichment cultures (SRB, APB, and GANB) were verified at the distinct sampling points (beta diversity). For this, alpha diversity data was obtained using PAST 4.08 [54]. These data were correlated with the relative abundances of the genera in each sample, also using the same program, through canonical correspondence analysis (CCA). The Shannon index (Shannon_H) was used to determine the attribute of diversity (richness and abundance), the Simpson index (Dominance_D) was used to assess dominance, and the Equitability Index was utilized to measure the uniformity of distribution of ASVs between groups [55]. In beta diversity, the correlations of information related to the distribution patterns of microbial genera within the ecosystem were evaluated based on Shannon diversity, dominance, and equitability indices.

For the analysis of shareability and uniqueness of microbial genera among the samples, the UpSetR package (version 1.4.0) in R was used [56]. To plot the bar graph by phylum, a threshold of 0.5% was used, and those below 0.5% were merged into a same category named “Others”. For the genus level, a threshold greater than or equal to 1.0% was also used, and all genera below 1.0% were described as “Others”.

A statistical analysis of the association between genera and samples was performed (PW_1.00 m, PW_2.75 m, SRB_1.00 m, SRB_2.75 m, APB_1.00 m, APB_2.75 m, GANB_1.00 m, and GANB_2.75 m), represented in a network, so that the network shows only strong associations based on absolute abundances. The analysis was calculated using the indicspecies package version 1.7.6 [57] with the multipatt and the point biserial correlation coefficient (r.g.) as a function for calculating the strength of association (SA) between genera and samples. The significance cut-off point for the phi coefficient was set at *p* < 0.05. The association network was built in Cytoscape, version 3.5.1 [58] using the “edge-weighted spring embedded” layout. The size of the nodes (node degree) represents the logarithm of the absolute abundance of the genera. The length of the edges represents the strength of association: the shorter the edge length, the stronger the association between the genera and the samples.

Principal Component Analysis (PCA) was used to explore all data in an integrative way. PCA summarizes information from a very high number of original variables into a small set of statistical variables. In our study, the inter-relationships between sets of microbial genera were analyzed using the first two principal components (PC1 and PC2). The data were initially submitted to the Hellinger transformation and, later, analyzed using PCA and visualized using two-dimensional scatter plots using the packages ggplot2 v3.3.5 and factoextra v1.0.7 R [59,60]. Through the distance between sampling points (tank height and cultures from environmental inocula), it was inferred whether the characteristics were similar or divergent. The length of the vectors indicates the region that has more (or less) microbial genera influencing the variability of the data, and the angle formed between the variables indicates the correlation between them. Exploratory analyses were also performed through clustering methods. Cluster analysis were performed and associated with a color matrix for better visualization of the results (only those with an abundance of 0.5% or more were displayed), using the R Heatmaply package [61,62].

### 2.8. Correlations of the Composition of Microbial Communities to the Physicochemical Parameters of PW

In order to infer the potential metabolic functions of the microbial genera detected by metabarcoding, physicochemical characterization of the PW samples was initially performed. For this, the biocorrosion diagnostic protocols for the examination of water and wastewater were used [63]. The analyses included pH, organic acids (lactate, acetate, propionate, formate, and butyrate), sulfate, soluble sulfides, iron, chlorides, salinity and thermal conductivity, and all these analyses were adapted according to [38,39], along with alkalinity, modified by Petrobras [40]. Subsequently, physicochemical data were analyzed, in an integrative way, with the abundances of microbial genera, using the canonical correspondence analysis (CCA).

## 3. Results

### 3.1. MPN Methods

Enrichment cultures from sampling points p_1.00 m and p_2.75 m showed microbial quantification of 10^5^ MPN/mL of SRB and GANB, with slightly higher growth of APB, 10^6^ MPN/mL (Appendix A). Regardless of the results of MPN quantification, all culture flasks that showed positive growth were processed, and their DNA was extracted.

### 3.2. Shareability and Uniqueness Patterns and Relative Abundance of Phyla and Genera Revealed by Metabarcoding in PW and Enrichments

The main patterns of the communities of both produced water (PW_1.00 m and PW_2.75 m) and enrichments (SRB_1.00 m and SRB_2.75 m; APB_1.00 m and APB_2.75 m; and GANB_1.00 m and GANB_2.75 m) are depicted Figure 1.

PW_1.00 m and PW_2.75 m samples showed the highest number of unique gener compared to the other samples, with 45. Amongst the enrichment cultures, the SRB group was the most diverse, displaying 20 genera shared between them (SRB1.00 and SRB_2.75) and with the PW samples (PW_1.00 m and PW_2.75 m) (Figure 1A and Appendix A).

We observed that the phyla Halobacterota (6.68–7.72%) and Cloacimonadota (16.51–19.31%) were identified with a relative abundance value higher than 0.5% only in samples PW_1.00 m and PW_2.75 m (Figure 1B). Phylum Halobacterota was represented by methanogenic archaea, with a higher abundance of the genus *Methanothrix* (Figure 1C). In contrast, the phylum Cloacimonadota was represented by only one taxon described as Cloacimonadaceae LNR A2-18 (Figure 1C). The phylum Proteobacteria was significant only in samples of SRB_1.00 m and 2.75 m (Figure 1B), with a higher abundance of the genus *Marinobacterium* (16.04–43.82%). In contrast, unclassified organisms were more abundant in samples PW_1.00 m (22.70%), PW_2.75 m (21.18%), SRB_1.00 m (18.50%), SRB_2.75 m (11.39%), and GANB_1.00 m (11.47%).

There were 19 genera shared by metabarcoding in PW and the corresponding enrichments (Figure 1A). The phylum Desulfobacterota was highlighted (Figure 1B), and the genus *Desulfovibrio* displayed relative abundances above 1.9% in all samples (Figure 1C). It is worth noting that the enrichment SRB_2.75 m showed a higher relative abundance of *Desulfovibrio* (32.84%) compared to the other samples. However, in SRB_1.00 m, the relative abundance of *Desulfovibrio* was notably lower (1.9%) compared to other samples. 

### 3.3. Microbial Diversity of PW and Enrichments

Alpha diversity indices (diversity, dominance, and equitability indices) and Canonical Correspondence Analysis (CCA) are depicted in Figure 2A,B, respectively.

The highest microbial diversity corresponded to PW samples (PW_1.00 m and PW_2.75 m) (Figure 2A,B). A significant pattern was detected in both PW_1.00 m and PW_2.75 m samples, which showed a higher Shannon diversity index and equitability.compared to cultured samples (Figure 2A), except for the SRB_1.00 m and SRB_2.75 m samples, which did not present statistical differences with the PW_1.00 m and PW_2.75 m samples regarding equitability, (Appendix A). On the other hand, the dominance was low in PW samples (PW_1.00 m and PW_2.75 m). Moreover, GANB (1.00 m and 2.75 m) enrichments exhibited a higher dominance, compared to PW samples (Appendix A).

CCA analysis of the PW and enrichments explained 100% of the cumulative variance of the correlation between the diversity indices and the distribution of communities of the different sample types at distinct heights (Figure 2B). The vectors referring to diversity (Shannon index) and equitability indices share the same quadrant (a) and are close. These samples contained a group of methanogenic archaea that were not identified in the enrichments. The vector referring to dominance is located in the (c) quadrant, opposite to the diversity and equitability indices, with a higher dominance in GANB samples, especially of *Dethiosulfovibrio* (46.25–48.04%).

#### 3.3.1. Statistical Evaluation of Patterns among Microbial Communities Directly Retrieved from PW and Enrichment Cultures

Statistical analyses describing patterns of microbial communities obtained from both PW (PW_1.00 and PW_2.75) and corresponding cultures (SRB, APB and GANB) from PW samples, at heights of 1.00 m and 2.75 m, are depicted in Figure 3A,B. Specifically, in Figure 3A, the association analysis between microbial genera detected in produced water (PW) and cultures (SRB, APB, and GANB is displayed. The interaction network exhibited in Figure 3A comprises four distinct groups of samples: (1) PW (green), (2) SRB (pink), (3) APB (yellow), and (4) GANB (blue). Group 1 (PW) presentes two well-defined branches, representing the largest group of microorganisms. Groups 2 (SRB) and 3 (APB) exhibit three branches, and group 4 (GANB) displays two distinct branches (Figure 3A). Several microbial genera were significantly and exclusively associated with to both produced water and cultured samples. The highest number of exclusively associated genera (*n =* 65) was observed in the PW samples, followed by the cultured sample (SRB *n* = 15).APB samples showed only two genera statistically and exclusively associated, while GANB samples did not display any significant or exclusive associated genera (Figure 3A).

PW samples were associated with SRB and APB groups (Figure 3A). The interaction with the SRB group was performed through 12 nodes, with emphasis on *Desulfoplanes* (SA = 0.81 and *p* = 0.0002). In turn, the interaction with the APB group was through three genera of similar intensities: *Fusibacter* (SA = 0.68 and *p* = 0.0031), *Halanaerobium* (SA = 0.61 *p* = 0.012), and *Curvibacter* (SA = 0.69 *p* = 0.0047). The association between the APB and GANB groups was represented only by *Dethiosulfovibrio* (SA = 0.76 *p* = 0.0007), while *Halarcobacter* represented the association between the GANB and SRB groups (SA = 0.7, *p* = 0.0027).

Principal component analysis (PC1 and PC2) was performed to explore the inter-relationships between the variables and explain these variables in terms of dimensions (Figure 3B). PC1 and PC2 explained 74.50% of data variability: PC1 explained 53.40% and PC2 explained 21.10% of the data variability. The PW_1.00 m and PW_2.75 m samples were closer to each other, and the group variability encountered in those was similar and better explained by PC2 (Figure 3B). Similarly, the GANB 1.00 m and GANB_2.75 m cultures were also more related to each other, and the variability of microbial groups found in these samples was better explained by PC1, when compared to the other samples. APB samples also exhibited a similar behavior, with overlapped vectors. The different sample heights (1.00 m and 2.75 m) were not significant to explain the data variability.

#### 3.3.2. Cluster Analysis in Q and R Modes

In the cluster analysis visualized through a heat map (occurrence and abundance), the samples were separated into two main clusters (Figure 4). Cluster A was divided into two subclusters: A1 composed of samples GANB_1.00 m and GANB_2.75 m, and A2, composed of samples APB_1.00 m and APB_2.75 m. Cluster B was divided into two subclusters (B1 and B2). Subcluster B1 was composed of samples PW_1.00 m and PW_2.75 m. In turn, subcluster B2 was subdivided into two less inclusive subclusters, B2.1 and B2.2. Subcluster B2.1 was constituted by the sample SRB_1.00 m and subcluster B2.2 by the sample SRB_2.75 m (Figure 4). 

The groups were formed according to sample type: cluster A (GANB_1.00 m and GANB_2.75 m), and (APB_1.00 m and APB_2.75 m); and cluster B (PW_1.00 m and PW_2.75 m) and (SRB_1.00 m and SRB_2.75 m). In cluster A, dominance of the sulfate and thiosulfate-reducing genera, *Desulfovibrio* and *Dethiosulfovibrio*, was observed. The difference between these subclusters is directly related to the genus *Halarcobacter* (GANB_1.00 m and GANB_2.75 m) and the APB *Halanaerobium* and *Fusibacter* (APB_1.00 m and APB_2.75 m. In cluster B, the produced water samples PW_1.00 m and PW_2.75 m) showed a very similar profile, with high abundance of the thiosulfate-reducing genus *Dethiosufatibacter*, with relative abundances of 25.49% and 23.95%, respectively. In general, the PW samples exhibited a more homogeneous profile of genera and relative abundances above 0.5% (Figure 4).

However, the SRB cultures displayed less similar the microbial profiles between SRB_1.00 m and SRB_2.75 m when compared to other sample groups. A great discrepancy was observed in the relative abundances of the SRB *Desulfovibrio*, with 32.84% in the SRB_2.75 m sample and 1.98% in the SRB_1.00 m sample. Another contrasting pattern between distinct SRB samples by height occurred for *Marinobacterium*, with 43.82% in the SRB_1.00 m sample, and 16.04% in the SRB_2.75 m sample, as well as for the genus *Halarcobacter*, with 15.50% in the SRB_2.75 m sample, and 0.36% in the SRB_1.00 m sample (Figure 4).

### 3.4. Physicochemical Characterization

#### 3.4.1. Physicochemical Features of PW

Physicochemical conditions displayed different patterns when the evaluation was performed on samples collected at different heights in the tank (Table 1). Point p_2.75 showed a lower pH than point p_1.00. Conversely, butyrate and acetate concentrations were higher at p_2.75. The short chain organic acids (lactate, propionate, and formate) exhibited values below the detection limit. Moreover, anion analyses indicated similar results at both points, except for sulfate, which showed a higher concentration at 1.00 m compared to 2.75 m. 

#### 3.4.2. Associations between the Physicochemical Features and the Microorganisms Present in PW Environmental Samples

Associations were evaluated between the microbial genera belonging to physiological groups, such as APB, SRB, NRB, TRB, S0RB, IRB, MPA (methane producing archaea), hydrocarbon degradation bacteria (HD), and SYN (syntrophic) and the physicochemical features, such as acetate (Ac), alkalinity (Alc), iron (Fe), chloride (Cl^−^), butyrate (But), turbidity (Turb), soluble sulfides (S^−^), salinity (Sal), pH, sulfate (SO_4_^2−^), dissolved solids (DS),and electrical conductivity (EC) of PW samples (p_1.00 and p_2.75) (Figure 5). CCA analysis of environmental PW samples (PW 1.00 and PW 2.75) explained 100% of the cumulative variance between environmental factors and the distribution of microbial communities’ present at both heights (p_1.00 and p_2.75).

At the height of 1.00 m, a higher abundance of MPA, HD, IRB, and synthophic (SYN) prokaryotes was retrieved (Figure 5), with a joint location of groups S_0_RB, TRB and NRB, in the center of the graph. At the height of 2.75 m, a different microbial profile was observed with a higher amount of fermentative and sulfidogenic organisms (APB and SRB groups) (Figure 5).

## 4. Discussion

Our study evaluated the microbiota of environmental PW samples (PW) and the corresponding enrichment culture obtained in media for specific microbial groups (SRB, APB, and GANB). These samples were collected from two heights (1.00 and 2.75 m) of a drainage tank that receives fluids from other tanks located in a distribution terminal. Diversity (taxonomic composition, richness, abundance, evenness, and dominance patterns) as well as multivariate ordination and cluster analyses were performed using 16S rRNA metabarcoding both original produced water and enrichment cultures. Additionally, the physicochemical features of PW were also evaluated in order to associate them with the richness and abundance of the genera present in this environment and, thus, infer the putative metabolism of the studied microbial groups.

In this study, we observed that the growth of GANB was similar to those in SRB and APB cultures. Cultivation of GANB includes the growth of anaerobic organisms that use organic compounds as a carbon and energy source. This group can be considered broad and heterogeneous [64], and it is used as an index of the total contamination by anaerobic microbes in a system since they can promote the establishment of corrosive groups such as SRB to the metal surface [65]. On the other hand, diversity analysis indicated that the culturing of GANB exhibited a higher dominance index, compared to PW and other cultured samples (SRB and APB). Regardless of the culture media used, there was an evident strong selection process caused by the specificity of the culture medium and/or by the incubation conditions, such as temperature and anoxia [66].

High diversity observed in the PW samples (1.00 and 2.75 m) can be attributed to different microbial groups, which do not necessarily contribute to MIC since the unique genera identified from environmental samples displayed an abundance lower than 0.2% and are, therefore, quite rare. Nonetheless, this high diversity in environmental samples can be attributed to the more abundant organisms in the PW samples, such as the methanogenic archaea (6.23%), some of which are capable of causing biocorrosion [67], as well as unclassified microorganisms (22.70%). Conlette et al. [67] evaluated the effects of temperature and pH on corrosion rates and methane production using metal coupons. These authors reported that methane production was strongly correlated with corrosion rates and moderately correlated with pH. Despite not being properly characterized, these microorganisms have already been described in the literature, such as *Cloacimonadaceae* LNR A2-18, which is not culturable and about which there is no available information besides its 16S rRNA gene sequence [68]. According to Lu et al. [69], the non-culturable *Cloacimonadaceae* LNR A2-18 may play important roles in sites where anaerobiosis predominates.

Only five unique genera were detected in cultured samples, which are not known to be associated with MIC (*Brevundimonas*, *Enterococcus*, *Irregularibacter*, *Lactococcus*, and *Oenococcus*) with abundances lower than 0.02%. In contrast, environmental PW samples exhibited almost 10 times more unique genera, totaling 46, with relative abundances lower than 0.15%, besides unclassified organisms (Appendix A). This pattern is directly linked to a higher dominance of some microbial groups in cultures due to the strong selection caused by the culture media [66]. Additionally, the physicochemical and nutritional conditions favored the selection of organisms of interest [66,70]. Therefore, microorganisms less adapted to those specific conditions are excluded [66].

Only two genera, *Dethiosulfovibrio* and *Halarcobacter*, occurred in GANB medium, and they were significantly correlated with APB and SRB, but with no significant correlation with environmental PW samples. This may indicate that the GANB culture medium is not efficient at detecting anaerobic microbial diversity, as it was designed to do, probably due to the nutrient deficiencies in the culture medium of the present study, which lacked vitamins and residual metals. Sakamoto et al. [25] compared two culture media with different carbon/sulfate ratios for developing microbial groups present in UASB (Upflow Anaerobic Sludge Blanket) reactor sludge. These authors concluded that the nutritional conditions of the medium influenced the development of the groups and that the presence of vitamins and residual metal solutions favored the development of the group of general anaerobic heterotrophic bacteria (GANB) [25]. In our study, we observed significant and positive associations between uncultured PW samples and enrichment cultures. Nevertheless, Sterflinger et al. [71] question the range of organisms detected in uncultured samples. We emphasize that, in our study, it was possible to identify the presence of microorganisms previously characterized as involved in biocorrosion processes that were not identified in cultured samples, such as methanogenic archaea. However, it is worth noting that the 16S rRNA gene-based metabarcoding approach only provides insights into the taxonomic composition and community structure, not the functionality of the community [72]. While there are bioinformatics techniques that can be used to estimate metabolic activity, these are only predicted based on physicochemical characterization of PW samples and may not represent a true indication of activity [10].

*Desulfovibrio* was identified in all PW and enrichment cultures samples and displays an important function in the environment since all the species are SRB, which are reference microorganisms in biocorrosion monitoring in the oil industry [70,73]. However, a high abundance of methanogenic archaea was also observed in environmental PW samples, which may characterize a syntrophic behavior between these two microbial groups [74].

The relative abundance of the genus *Desulfovibrio* in enrichment cultures obtained from PW was increased in this group, since the culture media maximize growth. This suggests that a specific medium does not exclude the development of other groups [75] since, in our study, APB exhibited high abundances (15.75–28.52%) of *Desulfovibrio*, which is a genus of SRB. The strategy that can be used in this situation is the dilution of the inoculum, which eliminates “undesirable” fast-growing species in small quantities, allowing the development of slow-growing species, which are the most abundant organisms in the community [75]. On the other hand, a relative abundance of *Desulfovibrio* of only 1.98% was observed in the SRB_1.00 m, which is an abundance value lower than that of PW samples. Hence, selective media do not guarantee the full development of genera of the same microbial group of interest [75].

Sample features can also significantly influence the abundance of diagnostic genera [76]. For instance, in the our study, in PW collected at 1.00 m (near the bottom of the tank) and subsequently cultured in SRB medium, *Desulfovibrio* showed an abundance of 1.98%, whereas in PW collected at 2.75 m (near the oil/water interface of the tank) also cultured in SRB medium, *Desulfovibrio* abundance was more than 16 times higher (32.84%). CCA analysis retrieved the associations between the physicochemical features and the microbial genera present in the uncultured PW samples. Altogether, these retrieved patterns suggests that anaerobic digestion occurred, with distinct phases at heights of 2.75 m and 1.00 m.

The presence of APB, in conjunction with the lower pH and the higher concentrations of organic acids (acetate and butyrate) at 2.75 m, when compared to 1.00 m, indicates acidogenesis. APB metabolize the products generated in hydrolysis (the first step of the anaerobic digestion process) and secrete compounds such as organic acids, acetate, short-chain volatile fatty acids (SCFA), alcohols and ketones [77,78,79]. In acetogenesis, most acids, such as butyric acid, propionic acid, and ethanol, are oxidized into acetate, which can be used in methanogenesis and sulfidogenesis [77,80].

PW samples (PW_1.00 m) were collected closer to the bottom of the tank, suggesting lower nutrient availability in comparison with PW samples (PW_2.75 m) collected closer to the oil/water interface. In the PW_1.00 m samples, a dominance of syntrophic, methanogens and hydrocarbon degraders bacteria was detected, where a neutral pH (7.0) was observed, a higher value compared to the lower acidic pH (6.0) detected in the PW_2.75 m samples. This is, consistent with the higher concentration of organic acids and the higher abundance of methanogens, as they are sensitive to lower pH. The availability of high concentrations of sulfate can promote another anaerobic process, the sulfidogenesis [81]. In this stage, sulfate is reduced to H_2_S by SRB and SRA [77,82,83].

Besides the fermentative metabolism observed in our study, the abundance of the SRB group may suggest a syntrophic relationship between the aforementioned microbial groups. As APB partially metabolize organic compounds, SRB use fermentation products as electron donors, especially in the case of acetate and butyrate [84]. These volatile acids and propionate are the most important components in the process [84]. Nevertheless, the presence of sulfate in the SRB medium suggests that the preference is for sulfate rather than biological electron acceptors such as methanogens. Nonetheless, the high abundance of methanogenic archaea implies that the SRB are engaged in syntrophic metabolism, indicating a possible occurrence of methanogenesis at the depth of 1.00 m. Certain genera of sulfate reducers are capable of syntrophically breaking down partially oxidized organic substances, such as lactate, ethanol, and propionate, through hydrogen (electron) transfer between species, in association with hydrogenotrophic methanogens [74].

The TRB were the most abundant group, with a dominance of the genus *Dethiosulfatibacter* (23.95–25.49%), which was not influenced by physicochemical features, since it displayed equivalent abundances in both PW samples collected at heights of 1.00 m and 2.75 m. In our study, we did not confirm if the abundance of thiosulfate reducers is related to the presence of thiosulfate in the medium because this analysis was not performed. Nevertheless, we could infer that must have significant concentrations of thiosulfate at both heights, which may be capable of causing biocorrosion in the medium [85]. According to Atkinson and Richards [85], thiosulfate can be produced by the chemical oxidation of H_2_S with O_2_. The authors demonstrated that thiosulfate is produced through the electro-chemical dissolution of MnS in stainless steel during the onset of pitting corrosion.

The presence of anaerobic digestion, jointly with a high abundance of methanogenic archaea, suggest the occurrence of methanogenesis as well as there may be a positive and significant correlation between methane production and metallic corrosion [67].

## 5. Conclusions

Our findings showed that the metabarcoding study of PW samples, combined with physicochemical characterization of those samples is more representative in monitoring microbial groups in oil industry facilities, as compared to the MPN method, which is based on culturing. This makes it possible to obtain a much more comprehensive and accurate picture of the microbial communities occurring in the different environments of the oil industry. Therefore, the use of metabarcoding coupled with physicochemical characterization may promote more adequate and efficient microbiological monitoring parameters to control biocorrosive processes in petroleum industry facilities.

## Figures and Tables

**Figure 1 microorganisms-11-00846-f001:**
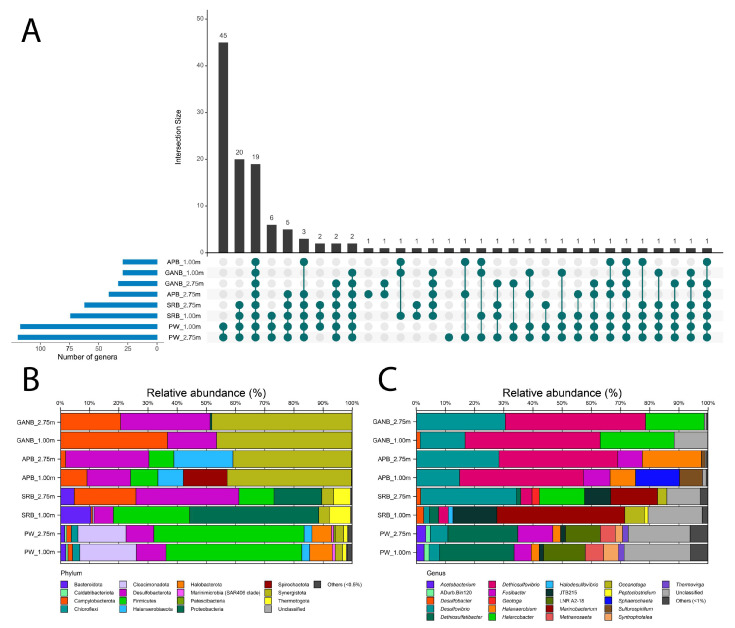
Shareability, uniqueness, and relative abundance (%) of identified microbial groups in metabarcoding PW samples (PW) at heights 1.00 m and 2.75 m and in, SRB, APB, and from a drainage tank. (**A**) Black vertical bars indicate the number of shared and unique genera among samples marked with green balls. Blue horizontal bars indicate the number of genera in each sample. (**B**) Bar graph. Each color indicates a different phylum, and the proportions indicate the abundance of each phylum in the different samples, at different heights. (**C**) Bar graph. Each color indicates a different genus, and the proportions indicate the abundance of each genus in the different samples, at different heights.

**Figure 2 microorganisms-11-00846-f002:**
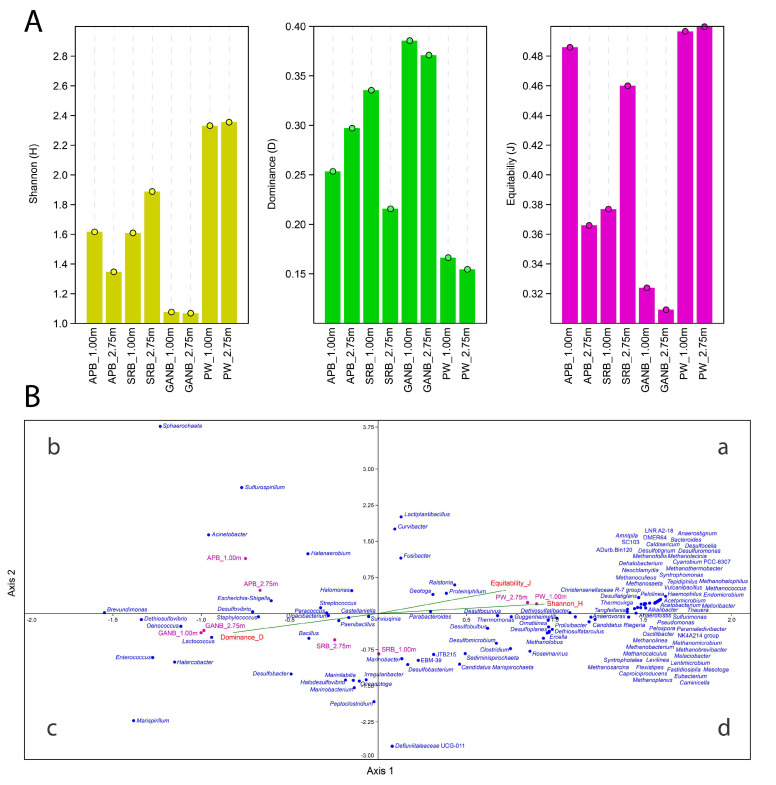
Alpha diversity indices and Canonical Correspondence Analysis (CCA) of microorganisms identified by metabarcoding in PW samples (PW) at heights 1.00 m and 2.75 m and, SRB, APB, and GANB enrichments at heights of 1.00 m and 2.75 m of a drainage tank (**A**) Vertical axis indicates the values of diversity indices, and the x axis indicates the analyzed samples (**B**) CCA: Pink circles indicate the sample type. The elements in red indicate the analyzed community ecological indices. Blue circles indicate the microbial genera that were detected in PW and enrichment cultures at different heights. Green vectors indicate the direction of each variable associated with the microbial genera of each sample type (a) quadrant I (b) quadrant II (c) quadrante III (d) quadrante IV.

**Figure 3 microorganisms-11-00846-f003:**
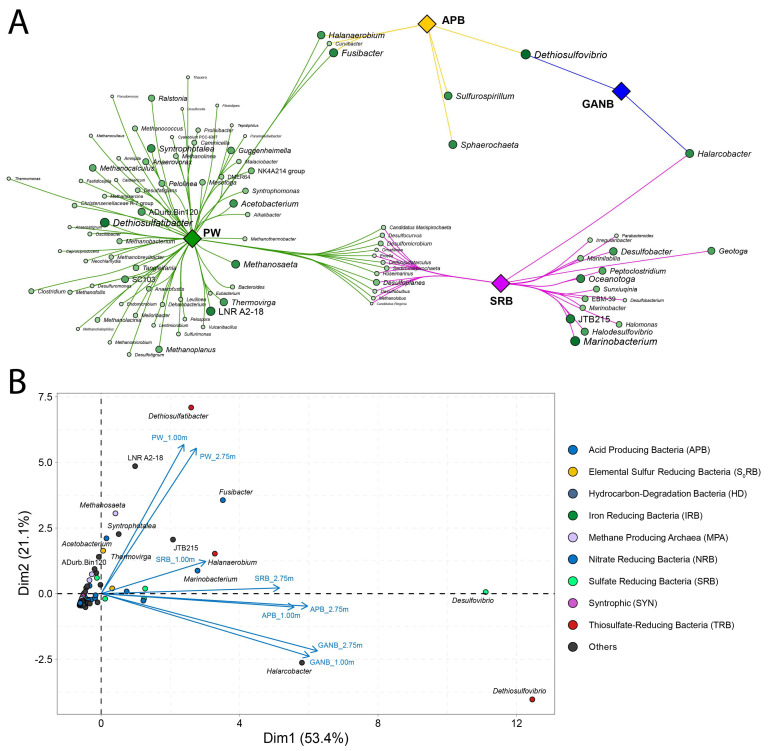
Association network and principal component analysis (PCA) of microorganisms identified by metabarcoding from PW samples (PW) at heights 1.00 m and 2.75 m and in enrichment cultures from PW samples, SRB, APB, and GANB at heights of 1.00 m and 2.75 m of a drainage tank (**A**) refers to association network by sample type and (**B**) refers to PCA analysis. In A, the diamonds indicate the nodes referring to the samples, which are differentiated by colors. The edges indicate the associations also represented in different colors depending on the sample type. The circles indicate the genera, with a longer diameter and higher color intensity indicating greater relative abundance. In B, the circles indicate the genus of microorganisms, and each color high lights a different group. Vectors refer to samples at different heights. PW: metabarcoding PW samples at heights 1.00 m and 2.75 m; and enrichment cultures from PW: SRB, APB and GANB at heights of 1.00 m and 2.75 m.

**Figure 4 microorganisms-11-00846-f004:**
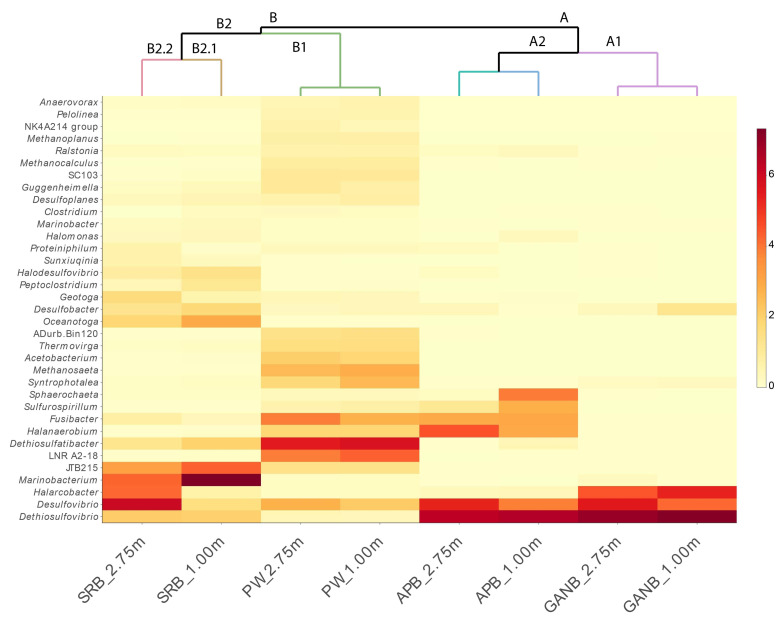
Dendrogram and color matrix of the abundances of microbial genera present in PW and enrichment cultures from a drainage tank. The columns indicate the formation of clusters and subclusters by sample, and the lines indicate the genera. Relative abundances are measured by the intensity of the color: the darker the color, the greater the relative abundance of a given genus.

**Figure 5 microorganisms-11-00846-f005:**
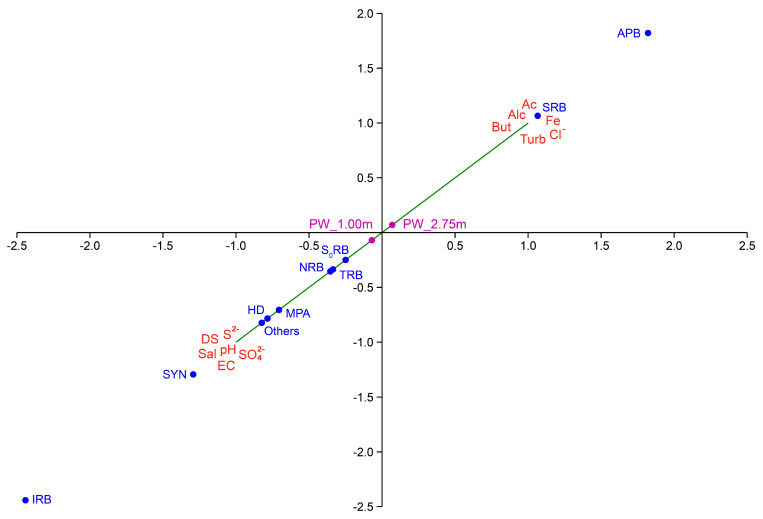
Canonical correspondence analysis (CCA) of the physicochemical features and abundance of microorganisms present in produced water (PW) samples from a drainage tank. The pink circles, indicate the sample type, the red abbreviations are the ions, organic acids, and physicochemical features of produced water (Ac = acetate; Alc = alkalinity; Fe = iron; Turb = turbidity; Cl^−^ = chloride; But = butyrate; S^2+^ = soluble sulfides; SO_4_^2−^ = sulfate; pH = hydrogen potential; EC = electrical conductivity; Sal = salinity and DS = dissolved solids). The blue circles indicate the ecological groups of microorganisms found in the environment, and the green vectors indicate the direction of each variable associated with the ecological groups of each sample.

**Table 1 microorganisms-11-00846-t001:** Physicochemical characterization of produced water (PW) samples collected from drainage tank at heights of 1.00 m and 2.75 m.

Feature	Unit	p_1.00	p_2.75
pH	**-**	7.00	6.00
Lactate	mg/L	n.d	n.d
Acetate	mg/L	970.00	1500.00
Propionate	mg/L	n.d	n.d
Formate	mg/L	n.d	n.d
Butyrate	mg/L	29.00	67.00
Sulfate (SO_4_^2−^)	mg/L	310.00	170.00
Soluble sulfides (S^2−^)	mg/L	57.80	54.90
Chloride (Cl^−^)	mg/L	2.60	2.80
Iron	mg/L	0.48	0.59
Alkalinity	meqs/L	25.50	28.70
Salinity	PPT	37.50	36.40
Electrical conductivity (EC)	Ms	65.00	60.71
Dissolved solids (DS)	PPT	31.55	30.25
Turbidity (Turb)	NTU	90.125	106.25

Sampling points: p_1.00 (1.00 m) e p_2.75 (2.75 m).

## Data Availability

The genome data are available in the NCBI repository under the Project number PRJNA928092 (https://www.ncbi.nlm.nih.gov/bioproject/?term=PRJNA928092—accessed on 28 January 2023).

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
