# Peer review of "Effective Biocorrosive Control in Oil Industry Facilities: 16S rRNA Gene Metabarcoding for Monitoring Microbial Communities in Produced Water"

_microorganisms, 2023, doi:10.3390/microorganisms11040846_

Round 1

Reviewer 1 Report

The study of de Cruz Ferraz Dutra et al. seems to be well designed and performed. However, some issues should be addressed.

1. The Discussion section should explore the significance of the results of the work. In some text fragments in the Discussion section, the Authors repeated the results. Please, correct it.   

2. The Authors wrote: “we have also indirectly inferred the probable ecological functions of these communities by complete reference genomes of the identified members from those communities.” Please explain how this step of the study was performed. I have not found any information about than in the Material and Methods Section. It should be completed. So, how did the Authors assigned the analyzed genera to the selected microbial groups such as SRB, APB, HD? Which approach was used.

3. Please cite the original source, which refers to  Postage  E medium, unless it is a modified version by Petrobrass. If this is the case,  please provide the composition. Please explain the TSI abbreviation (line 136) and provide the compositions of the media used to cultivate APB and GANB. 

4. What are the differences between acid-producing bacteria and general  anaerobic bacteria? Do the former group refer to fermentative bacteria and the latter refer to the total number of heterotrophic anaerobic bacteria?  Many anaerobes generate acids as end products of their metabolism. This is not clear; therefore, it must be specified. Please also provide the composition of the media for APB and GANB.

5. What kind of paired-end technology was applied? 2x250 bp or 2x300 bp?

6. There is an error with this link: https://github.com/LBMCF/pipeline-for- 161 amplicon-analysis. Please, correct it. What program was used to infer about ASV? It is clue of the analysis but it was not mentioned (DADA, USEARCH)? Did you remove chimera? If the unoise3 command was applied to remove chimera, it should be mentioned. I would suggest to indicate the subsequent steps of the pipeline used in this analysis and specify the programs used to perform each step, i.e. quality control of reads (FastQC), merging of the reads (USEARCH).

7.  I am quite surprised  that quantitative microbial analyses gave the same results for SRB and APB (Table 1).  It is also unusual  that the number of fermentative bacteria is higher than the total number of  heterotrophic anaerobic  bacteria (if my above assumption is correct). Please kindly provide SD or SE values .

8. I am not sure whether the discussed activity of methanogens can be attributed to the MIC. Some consume acetate or C1 compounds,  whereas others use CO2; therefore,  their activity should counteract the negative effects of other anaerobic bacteria. 

9. References should be cited according to the Journal’s style. That is: “References must be numbered in order of appearance in the text (including table captions and figure legends) and listed individually at the end of the manuscript. Please see https://www.mdpi.com/journal/microorganisms/instructions. Perhaps MDPI Stuff would help.

10. Please correct the caption of Figure2.

11. There is no statistical assessment of the results showed on Fig. 2. The Authors wrote that “a marked pattern was detected in both ENV 1.00 299 m and ENV 2.75 m samples, which displayed a higher Shannon diversity index and evenness as compared to all cultured samples (Figure 2A)”. Are these differences statistically significant? Please clarify: (302 line) “Moreover, GANB (1.00 m and 2.75 m) and SRB 1.00 m samples exhibited a higher dominance” Higher than?

12. Line 40: “associated with physicochemical characterization” of what?

13. Line 37-39: “There were also significant associations between metagenomic and cultured sample units, as well as significant enrichment (overrepresentation) of anaerobic metabolic pathways, which correlated to physicochemical features”. What do the Authors mean by “enrichment of anaerobic metabolic pathways”? The Authors did not analyze the pathways based on 16S amplicons or by any other methods? In my opinion, there this fragment of the statement is not precise; there is not proofs for that in the main body of the manuscript. Please clarify.

14. Please correct the title of the section 2.5.

15. What program was used to calculate the relative abundance? Please clarify.

16.Line 190: “Comparative analyses of PW sample’s amplicon metagenomics data and its corresponding cultured PW samples (SRB, APB, and GANB) were verified at the distinct sampling points.” What do the Authors mean by: “the comparative analyses were verified”?

17. Line 206: Please provide the reference for the indicspecies package.

18. Line 210” Please provide the reference for the Cytoscape.

19. Line 258 and 260 indicate the relative abundance not abundance.

20. Line 276: “There are 19 genera shared between the amplicon metagenomic and corresponding cultured PW samples”, in turn in the abstract it is written: „All samples had a core community composed of 16 distinct genera (…)”. Please verify.

21. Line 331: Please correct the name of bacteria: Dethiosulfovibrio.

22. Table 1. Please indicate if the results provided in Table 1 are from a single examination or are from replicates. If they are the mean values, please provide the SD. Please also clarify how this examination (replication/single measurement) were made in the “Material and Methods” section.

23. Line 408: Please explain MET abbreviation since there is no explanation in the text.

24. Line 411: It should be: in environmental PW samples.

25. Line 421: These features are not physical, they are mostly chemical ones. Please, correct : and physicochemical features.

26. Figure 5. There should be some kind of consistency in the presented data within the manuscript. First remark: in the main text the Authors wrote that HD abbreviation refers to hydrocarbon-degrading bacteria, while there is hydrocarbon degradation (HD) in the Figure 5. Please correct this error. Please also indicate in the main text not mentioned microbial groups that were analyzed, i.e. MPA, S etc.

27. Please explain this statement (line 471-472): In contrast, environmental samples exhibited almost 10 times more unique genera, totaling 46 (Figures 1A), with abundances lower than 0.15 %, besides unclassified organisms.” Figure 1 A shows that both environmental samples share 45 genera, which are unique for them (samples) compared to the other samples. Each environmental sample contains more than 100 genera. It means that it contains more less 50 unique genera. Please clarify.

28. Please correct this fragment (line 476) “(…) PW samples cultivated in GANB” the GANB is abbreviation for bacteria. Perhaps “cultivated in GANB medium” or (“GANB-supporting medium”) will be more appropriate.

29. Line 490: Figures 3A and 3B

30. Line 514: The Authors wrote: “anaerobic digestion metabolism was detected”. It is not true, please correct. The 16S amplicon sequencing does not provide any insight about the metabolic activity. The Authors did not show any proof for anaerobic digestion metabolism.

31. Line 569: Please clarify: “the use of these strategies”. In my opinion, the use of this strategy (16S rRNA amplicon sequencing)

32. Wrong citation: Machuca and Salgar-Chaparro: Complementary DNA/RNA-Based Profiling: Characterization of Corrosive Microbial Communities and Their Functional Profiles in an Oil Production Facility. Front. Microbiol. It should be Salgar-Chaparro and Machuca. Please, correct it.

33. Please cite the original paper, where the term amplicon metagenomica is used. It is not Salgar-Chaparro and Machuca (2019).

34. Line 154: Wrong citation. It should be Caporaso et al. (2010)

 https://doi.org/10.1073/pnas.1000080107. Please correct it.

35. Figure 1B and lines 202-203: there is some mistake. It is written that a threshold of 1.0% was used and phyla <1% were merged to category “others”, but in Figure 1B the category “Others <0.5%” is presented. Please, correct it.

36. Please explain how did the Figure 1C was constructed. What was presented in this Figure? Are every genera identified in the analyzed samples, presented on this Figure? Or, only top most abundant genera was presented? I would like to see the relative abundance table of the genera. Could the Authors provide such table as a response to the reviewer.  According the data in Figure 1A, there is more than 100 genera in each of environmental samples. More or less 15 genera in ENV_1.00 and ENV_2.75, were presented on the Figure 1C. Does it mean that more or less 80 genera exhibited relative abundance < 1%? Please explain.  

37. Please, add some statements of the limitations of 16S amplicon sequencing in the context of activity of microbial community and corrosion. It should be emphasis that this approach gives only the information about the taxonomic composition and community structure but not about the in situ activity of the community. Of course, there are several bioinformatic approaches used to infer about the metabolic activity of the analyzed community. Based on this approach, we can predict the activity, but it is only predictions not the real view of this activity.  

38. Line 564: it should be rather: 16S rRNA amplicon metagenomics study or 16S rRNA amplicon sequencing.

39. Please correct in the appropriate fragments, that physicochemical characterization or features refer to samples (physicochemical characterization of produced water (samples)).

Author Response

Dear reviewer,

I wanted to take a moment to express my sincere gratitude for taking the time to read and review my manuscript for Microorganisms. Your thorough review and thoughtful comments were incredibly helpful and will undoubtedly improve the quality of the final publication. Your willingness to share your expertise and provide constructive feedback is truly appreciated. Your insightful suggestions have challenged me to think more deeply about my research and have helped me to refine my work. Thank you again for your time and effort in reviewing my manuscript. Your contributions have been invaluable, and I am grateful for your support.

Reviewer 2 Report

The authors did a good job, I can see that the quality of the manuscript is generally good. Notwithstanding, I have these few observations.

1. The way the authors were writing their percentages seems not good. There should be no space between the figure and unit when writing percentage, e.g. 20%. This should be corrected all through the document.

2. Lines 122-123: The authors mentioned some physicochemical characterizations carried out but provided no details about how they were done, or the equipment used in doing them.

3. Lines 124 & 135: What were the specific culture media used.

Author Response

Dear reviewer,

I wanted to take a moment to express my heartfelt gratitude for taking the time to read and review my manuscript for [nome da revista científica]. Your constructive feedback and thoughtful comments have been immensely helpful in improving the quality of my work.

Reviewer 3 Report

See attached file.

Author Response

Dear reviewer,

I wanted to express my sincere appreciation for taking the time to read and review my manuscript for Microorganisms. Your thorough review and positive feedback were very encouraging and have made all the hard work worthwhile. Once again, thank you for your time and effort in reviewing my manuscript. Your support and encouragement have been invaluable to me.

Round 2

Reviewer 1 Report

Dear Authors, 

thank you for your revised version of manuscript. All the remarks were addressed. I read the manuscript carrefully once again. I would like to ask the Authors to correct the English language of the manuscript since there are some errors.

Line 125-127: Please correct the grammar of this fragment. In the current form, it is not understandable. Please, take care about the proper formula of the compounds (S2-, Cl- etc.). Please also, add information about the equipment such as manufacture (ion exchange chromatograph, pHmeter etc.). How was the iron content analysed? Is it Fe3+ or Fe2+? Please clarify this in the text.

“(…) physicochemical characterization (i)Lactate, acetate, propionate, formate, butyrate, sulfate (SO₄²⁻), soluble  sulfides (S2-), and chloride (Cl-) can be analyzed through ion exchange chromatography.  [38,39], and alkalinity by potentiometric titration and iron [40]. The analysis of pH and  temperature were evaluated using a pH meter (ii) culturing by inoculation in distinct flasks  containing specific culture media for the growth of SRB, APB, and GANB; (iii) filtering with the use of membranes with 0.22 μm diameter pores using vacuum pumps for microbial cell retention and subsequent extraction of metagenomic DNA.”

For example: (…)(i) physicochemical characterization (lactate, acetate, propionate, formate, butyrate, sulfate (SO₄²⁻), soluble sulfides (S2-), and chloride (Cl-)), (ii) culturing by inoculation in distinct flasks  containing specific culture media for the growth of SRB, APB, and GANB; (iii) filtering with the use of membranes with 0.22 μm diameter pores using vacuum pumps for microbial cell retention and subsequent extraction of metagenomic DNA. The mentioned fatty acid and ions were analyzed using ion exchange chromatography as described in [38,39], while alkalinity by potentiometric titration and iron [40]. The analysis of pH and temperature were evaluated using a pH-meter.  

Line 178: It should be: 2.5. 16S rRNA gene amplicon sequencing

Line 250: Please remove “R” or add “in” (in R).

Line 295: It should be Supplementary

Line 339: GANB samples exhibited a higher dominance than…? It should be explained since the Authors made the comparisons of GANB-ENV, SRB-GANB and GANB-APB.

Line 449: HD or (HDB) abbreviation should refer to hydrocarbon degrading bacteria since The Authors mentioned about the bacteria and archaea in the previous line, they did not mention about the process (hydrocarbon degradation). In my opinion is better to use hydrocarbon-degrading bacteria (HD) than hydrocarbon degradation. The Figure 5 should be also corrected. Please, change hydrocarbon degradation to hydrocarbon degrading bacteria.  

Figura 5 should be corrected (Figure 5).

Line 508-509: Please correct:  “Only five unique genera were detected in cultured samples, which are not known to contribute to associated with MIC”. There is some error: to contribute to associated

Line 512: Please check the orthography “Supplementary” 

Author Response

Dear reviewer,

I am writing to express my sincere gratitude for your willingness to review my article "Effective biocorrosive control in oil industry facilities: 16s rRNA amplicon metagenomic approach for monitoring microbial communities in produced water". Your contribution was essential to improve the quality of my work, and I greatly appreciate all the time and effort you have dedicated to this project. Your willingness to help, and provide constructive feedback, knowledge, and professionalism were remarkable, and I truly value all your work in my article.

Sincerely,
